# Depletion of VGLL4 Causes Perinatal Lethality without Affecting Myocardial Development

**DOI:** 10.3390/cells11182832

**Published:** 2022-09-10

**Authors:** Caroline Sheldon, Aaron Farley, Qing Ma, William T. Pu, Zhiqiang Lin

**Affiliations:** 1Masonic Medical Research Institute, 2150 Bleecker St., Utica, NY 13501, USA; 2Biology and Chemistry Department, SUNY Polytechnic Institute, 100 Seymour Rd., Utica, NY 13502, USA; 3Department of Cardiology, Boston Children’s Hospital, 300 Longwood Avenue, Boston, MA 02115, USA

**Keywords:** Vestigial-like 4, tandem Tondu domains (TDUs), cardiomyocyte, myocardium, development

## Abstract

Congenital heart disease is one of the leading causes of pediatric morbidity and mortality, thus highlighting the importance of deciphering the molecular mechanisms that control heart development. As the terminal transcriptional effectors of the Hippo–YAP pathway, YAP and TEAD1 form a transcriptional complex that regulates the target gene expression and depletes either of these two genes in cardiomyocytes, thus resulting in cardiac hypoplasia. Vestigial-like 4 (VGLL4) is a transcriptional co-factor that interacts with TEAD and suppresses the YAP/TEAD complex by competing against YAP for TEAD binding. To understand the VGLL4 function in the heart, we generated two VGLL4 loss-of-function mouse lines: a germline *Vgll4* depletion allele and a cardiomyocyte-specific *Vgll4* depletion allele. The whole-body deletion of *Vgll4* caused defective embryo development and perinatal lethality. The analysis of the embryos at day 16.5 revealed that *Vgll4* knockout embryos had reduced body size, malformed tricuspid valves, and normal myocardium. Few whole-body *Vgll4* knockout pups could survive up to 10 days, and none of them showed body weight gain. In contrast to the whole-body *Vgll4* knockout mutants, cardiomyocyte-specific *Vgll4* knockout mice had no noticeable heart growth defects and had normal heart function. In summary, our data suggest that VGLL4 is required for embryo development but dispensable for myocardial growth.

## 1. Introduction

Congenital heart disease occurs in around 1% of live births [1], and it is the leading cause of infant mortality [2]. Understanding the molecular mechanisms controlling heart development will provide novel insights for diagnosing and preventing congenital heart diseases. The Hippo–YAP signaling cascade is a key pathway that regulates heart development [3], and disturbances in Hippo–YAP have been implicated in a range of heart diseases, including arrhythmogenic cardiomyopathy [4] and Duchenne’s muscular dystrophy [5,6].

The core canonical Hippo pathway in mammals comprises a serine/threonine kinase cascade consisting of sterile 20-like kinases MST1/2 (homologs of the Drosophila Hippo kinases), scaffold protein Salvador homolog 1 (SAV1), and LATS1/2 serine/threonine kinases. SAV1 directly interacts with and activates MST1/2, and the activated MST1/2 in turn phosphorylates and activates LATS1/2 [7]. YAP and its homolog TAZ are the terminal transcriptional effectors of the Hippo pathway, and LATS1/2 restrain the YAP/TAZ activity through directly phosphorylating YAP/TAZ on their conserved serine residues [8,9]. In mice, knocking out SAV1 in the fetal heart caused cardiac hyperplasia [10], and the inducible depletion of SAV1 in the postnatal or adult heart increased cardiomyocyte regeneration capacity [11]. Similarly, the cardiac-specific depletion of LATS1/2 in the adult heart also increased cardiomyocyte renewal activity [11]. These published data demonstrate that Hippo kinases impede cardiac regeneration capacity.

One of the effects of the LATS1/2 phosphorylation of YAP/TAZ is to retain these transcriptional effectors in the cytoplasm [8]. Once the Hippo signal is off, LATS1/2 no longer retain YAP/TAZ in the cytoplasm, and these two co-transcription factors translocate into the nucleus, where they interact with DNA through TEA domain (TEAD) family transcription factors (TEAD1-4 in mammals) and other transcriptional factors to potently stimulate proliferation and promote cell survival. Both YAP and TEAD1 are required for heart development, as reflected by the fact that depleting either of these two genes in fetal cardiomyocytes caused cardiac hypoplasia [12,13,14]. Furthermore, the depletion of YAP in endothelial cells impaired atrioventricular development, and consequently, most of the YAP mutants could not survive to the weaning stage [15]. These published data highlight the importance of YAP and TEAD1 in heart development and suggest that investigating YAP and TEAD1 modulators may be a productive strategy to dissect the molecular mechanisms underlying congenital heart disease.

Different from LATS1/2, Vestigial-like 4 (VGLL4) is another category of YAP suppressors, which does not interact with YAP and functions by competing against YAP for TEAD binding [16,17]. In mice, the overexpression of VGLL4 completely abolished YAP-induced liver overgrowth and tumorigenesis [18]. VGLL4 has two conserved tandem Tondu domains (TDUs) that mediate the VGLL4–TEADs interaction, and these TDUs are both essential and sufficient for blocking the YAP activity [19]. In addition to binding YAP, TEAD4 also forms a complex with transcription factor 4 (TCF4), one of the terminal effectors of the Wnt/β-catenin pathway, thus linking the Hippo–YAP and Wnt/β-catenin pathways on the transcriptional level [20]. Interestingly, VGLL4 blocks the formation of the TCF4–TEAD4 complex and applying a VGLL4–TDU peptide mitigates colorectal carcinoma growth in mice [20].

We previously studied the relationship between YAP, TEAD1, and VGLL4 in the mouse heart, showing that the major interaction partner of TEAD1 switches from YAP in the neonatal heart to VGLL4 in the adult heart. Overexpressing a K225R-mutated VGLL4 in the neonatal heart disrupted the TEAD/YAP interaction and caused cardiac hypoplasia and heart failure [21]. Nevertheless, it remains unclear whether the loss of VGLL4 is required for myocardium growth, and whether it controls adult heart homeostasis. In the current work, we generated two *Vgll4* loss-of-function mouse models to define VGLL4 function in the heart: one germline *Vgll4* mutant allele and one cardiomyocyte-specific *Vgll4* depletion mouse line. In both mouse lines, VGLL4 TDUs were targeted to abolish the VGLL4 function. Although the whole-body VGLL4 depletion resulted in small embryo size and perinatal lethality, the cardiomyocyte-specific depletion of VGLL4 did not cause heart growth defects. Our data, therefore, suggest that VGLL4 is required for embryo development but dispensable for myocardial development.

## 2. Materials and Methods

Appendix A provides expanded material and experimental procedures.

### 2.1. Mice

All animal procedures were approved by the Institutional Animal Care and Use Committee of Boston Children’s Hospital and the Masonic Medical Research Institute.

### 2.2. Gene Expression

The total RNA was isolated using the Trizol reagent (Thermo Fisher Scientific Inc., Waltham, MA, USA). For quantitative reverse transcription PCR (qRT-PCR), RNA was reverse-transcribed (superscript III, Thermo Fisher Scientific Inc., Waltham, MA, USA), and specific transcripts were measured using SYBR Green chemistry and normalized to *Gapdh*. The primer sequences are provided in Appendix A.

### 2.3. Statistics

Values are expressed as mean ± SD. Student’s *t*-test was used to test for statistical significance involving two groups.

## 3. Results

### 3.1. Vgll4 Gene Has Different Transcript Variants

In mice, the *Vgll4* gene has seven annotated transcript variants (NCBI) that encode five VGLL4 protein isoforms (Figure 1A, Appendix A). Among the VGLL4 protein isoforms, their N-termini are diversely encoded by one or two exons, and their C-termini are identically encoded by three common exons (Figure 1B). Since TDUs are encoded in the last exon (exon 5), all the *Vgll4* variant products have TDU domains (Figure 1B). Based on whether containing exon 2, we categorized the *Vgll4* transcript variants into two groups: (I) *Vgll4* variants containing exon 2 (variant 1 and 2; X1 and X2) and (II) *Vgll4* variants without exon 2 (X3 and X4).

One of the group I *Vgll4* variants (variant 1) has been validated in the mouse heart [22]. We investigated whether the mouse heart expresses group II *Vgll4* variants. Using *Vgll4* variant 1 as a positive control, we examined the expression of *Vgll4* X3 and X4 variants in 1-day-old (P1) and 28-day-old (P28) mouse hearts. In both P1 and P28 hearts, the expression of *Vgll4* X4 was minimum, and the expression of *Vgll4* X3 was significantly higher than that of *Vgll4* variant 1 (Figure 1C).

### 3.2. Whole-Body Deletion of VGLL4 Domains Results in Perinatal Lethality

The presence of multiple *Vgll4* transcript variants makes it hard to perform genetic VGLL4 loss-of-function studies. The published data suggest that VGLL4 functions through its tandem TDU domains [18,19], which are encoded in the last exon of all the *Vgll4* variants (Figure 1A). To completely disrupt VGLL4 function, we used Cas9-mediated genome editing technology to target the last exon of *Vgll4* [23]. We obtained a *Vgll4* mutant allele with two indel mutations in the last exon: one 34 bp deletion and one 12 bp deletion (Figure 2A; Appendix A). This 46-base-pair deletion, which was named d46, caused a frameshift, thereby disrupting the highly conserved TDU domains (Figure 2A).

*Vgll4*^d46/+^ (*Vhet*) intercrosses yielded *Vgll4*^d46/d46^ (*Vko*) embryos at a Mendelian ratio at E16.5 and E18.5, but at postnatal day 1 (P0), only 8 of the 83 genotyped pups had the *Vko* genotype (Figure 2B). The majority of the surviving *Vko* pups died at P0, and none survived past P10 (Figure 2C). At E16.5 and P0, *Vko* mutants were much smaller than littermate controls (Figure 2D,E). Only a few mutants survived to P5, and these pups showed severe growth retardation (Figure 2F,G). *Vgll4* deletion was confirmed by using qRT-PCR with two sets of *Vgll4* primers: one pair of primers anneal to the deleted region (TDU domains) and the other pair to the unaffected five-prime region. The TDU primer sets failed to detect signals in *Vko* transcripts; however, the primers amplifying *Vgll4* five-prime regions did not detect a difference between *Vhet* and *Vko*, suggesting that the indel mutations disrupt the TDU-coding region without affecting the stability of *Vgll4* mRNA. Together, these data show that VGLL4 is required for embryo development and normal postnatal growth.

### 3.3. Whole-Body Deletion of VGLL4 Does Not Affect Myocardial Development and Heart Function

We previously showed that overexpressing a mutated VGLL4 suppresses postnatal heart growth [21]. Therefore, we focused on testing whether the loss of VGLL4 impairs heart development. In contrast to the robust body weight difference, the heart weight did not show a significant difference between *Vhet* and *Vko* at P0 (Figure 3A). Consequently, the heart-to-body weight ratio of *Vko* pups was much higher than that of the *Vhet* controls (Figure 3B). The systolic heart function was similar between *Vhet* and *Vko* at P0 (Figure 3C), indicating that the perinatal lethality of *Vko* pups is not due to heart failure.

To define whether *Vgll4* d46 is a recessive mutation, we compared the heart weight, body weight, heart function, and *Vgll4* mRNA level between *Vhet* and wild-type (WT) control pups. At P0, all these four parameters showed no significant differences between *Vhet* and WT pups (Appendix A), suggesting that the loss of one *Vgll4* allele does not affect fetal development.

As the YAP/TEAD1 complex is essential for fetal CM proliferation [12], and VGLL4 is a suppressor of this complex [21], we investigated whether the inactivation of VGLL4 increases fetal CM proliferation. 5-ethynyl-2′-deoxyuridine (EdU), an analogue of thymidine, was used to label proliferating CMs at E16.5. As shown in Figure 3D,E, the CM EdU incorporation rate was not distinguishable between *Vhet* and *Vko*. Additionally, compared with *Vhet*, *Vko* had a similar left ventricle free wall and septum thickness (Figure 3F,G). In line with these observations, the expression of several cell proliferation genes, including *CyclinB1* (*CycB1*), *Cdk1*, and *Igf1r*, was not altered in the *Vko* hearts (Figure 3H). We further examined three well-known YAP/TEAD target genes: cellular communication network factor 2 (*Ccn2*, also known as *Ctgf*) [24], cellular communication network factor 1 (*Ccn1*, also known as *Cyr61*) [25], and baculoviral IAP repeat containing 2 (*Birc2*, an apoptosis inhibitor gene) [8]. Although *Ccn2* was upregulated in the *Vko* heart, *Ccn1* and *Birc2* showed no difference between *Vhet* and *Vko* (Figure 3I), suggesting that *Vgll4* partially suppresses the YAP/TEAD complex in the fetal heart.

These data show that VGLL4 is dispensable for myocardial growth and suggest that the perinatal lethality of *Vko* mutants is not due to heart dysfunction.

### 3.4. Whole-Body Deletion of VGLL4 Causes Tricuspid Valve Malformation

To determine whether the whole-body deletion of VGLL4 affects the heart structure, we prepared longitudinal sections with the P0 *Vko* hearts. *Vko* hearts had preserved morphology, with the exception of the tricuspid valve, which appeared dysmorphic (Appendix A). This observation led us to investigate whether the loss of VGLL4 impairs tricuspid valve development; the *Vko* embryo serial sections were then examined for tricuspid valve defects at E16.5. The four-chamber view of the heart sections displayed long and thin tricuspid valve leaflets in *Vhet*; however, the counterparts in *Vko* hearts were short and thick (Figure 4A; Appendix A). Due to technical issues, we were unable to determine the relative position (anterior or posterior) of the right ventricle side leaflets. Therefore, we used right ventricle side tricuspid valve leaflets (RVL) to represent these two leaflets. The measurement results of the RVL and septal leaflets (SLs) showed that *Vko* hearts had short RVLs and normal SLs (Figure 4B,C), confirming that the loss of VGLL4 impairs tricuspid valve development.

We further measured the tricuspid valve interstitial cell (VIC) proliferation rate by labeling the fetal heart with EdU for two hours. Compared with *Vhet*, the *Vko* embryos had a significantly lower EdU incorporation rate in the RV leaflets. Although the *Vko* septal leaflets also had lower EdU incorporation rates, the difference did not reach significance (Figure 4C,D).

### 3.5. Generation of a Vgll4 Flox Allele

Due to the fact that the *Vgll4* d46 mutants die perinatally, we were unable to examine whether VGLL4 regulates postnatal heart growth. To systematically study VGLL4 function in the postnatal heart, we made a *Vgll4* flox allele that targets the coding sequence of its last exon. The last exon (exon 5) of *Vgll4* encodes TDU domains and has a long untranslated region (UTR). To perform recombinant genome targeting, we chose one 1167 bp intron fragment and one 1074 bp three-prime UTR as the left and right recombination arm, respectively (Appendix A). In the *Vgll4* flox targeting vector, the protein-coding sequence of *Vgll4* exon 5 was flanked by LoxP sequences, and an FRT sequence-flanked neomycin resistance cassette was placed between the right LoxP site and the right recombination arm (Appendix A). To increase the genome targeting efficiency, we employed CRISPR-Cas9 technology to cause a lesion region on the *Vgll4*’s right recombination site. Correspondingly, for protecting the *Vgll4* targeting vector, a CGG > CAG mutation was introduced into the right recombination arm to destroy the short-guide RNA (sg_3) PAM sequence (Appendix A).

After mouse embryonic stem (ES) cell targeting, neomycin-resistant clones were screened with PCR (Appendix A), and positive clones with normal karyotypes were used for generating the *Vgll4* flox founder lines. The germ-line transmission of the *Vgll4* flox_neomycin (*Vgll4*^fl_Neo^) allele was successful. Since the presence of a selectable marker cassette may disrupt the expression of neighboring genes [26], we decided to remove the neomycin-resistant cassette from the *Vgll4*^fl_Neo^ mouse line, in which the neomycin selection cassette was flanked with FLP recombinase-recognized FRT sequences. *ROSA26-Flp* is a transgenic mouse line that has the FLP recombinase gene knocked into the *ROSA26* locus, thus constitutively expressing FLP in all tissues [27]. We then crossed *Vgll4*^fl_Neo^ with *ROSA26-Flp* to remove the neomycin resistance cassette (Figure 4A). We sequenced the floxed region of the *Vgll4* flox allele and confirmed the insertion of two LoxP fragments flanking the exon 5 coding sequence and the removal of the neomycin resistance cassette (Appendix A).

The homozygous *Vgll4* flox mice (*Vgll4^fl/fl^*) normally bred and had normal systolic heart function (Appendix A) and an unchanged heart-to-body weight ratio (Appendix A). The *Myh6-Cre* transgenic mouse line has been widely used for depleting floxed alleles in the postnatal heart [28]. To study the VGLL4 function in the postnatal heart, we crossed the *Vgll4* flox mice with the *Myh6-Cre* mice to generate a cardiac-specific *Vgll4* knockout mouse line (*Vgll4^cKO^*) (Figure 5A). The adult heart has multiple cell types, with CMs only accounting for ~35% of the cardiac cellular composition [29]. To validate the *Vgll4* depletion efficiency and avoid potential contaminations contributed by non-CMs, we dissociated *Vgll4^fl/fl^* and *Vgll4^cKO^* hearts and acquired purified CMs. The *Vgll4* exon 5 excision efficiency was tested with CM genomic DNA PCR, which showed that the floxed region was efficiently removed in *Vgll4^cKO^* CMs (Figure 5B). Consequently, *Vgll4* mRNA was substantially depleted in *Vgll4^cKO^* CMs (Figure 5C).

We further tried to examine the VGLL4 protein level; however, we failed to find a VGLL4 antibody that can specifically recognize VGLL4 in mouse heart lysate. To circumvent this obstacle, we used TEAD1 as bait to pull down VGLL4 from the adult heart lysate and used a custom VGLL4 antibody (see Appendix A) to detect VGLL4 in the TEAD1 pull-down products. In line with our previous results indicating that VGLL4 suppresses TEAD1 expression [21], the TEAD1 protein level was increased in the *Vgll4^cKO^* heart (Figure 5D). In the input heart lysate, VGLL4 was not detectable. In the TEAD1 pull-down product, two VGLL4 bands were detected. Furthermore, in the TEAD1 pull-down products, although the VGLL4 protein level was similar between *Vgll4^fl/fl^* and *Vgll4^cKO^* hearts, *Vgll4^cKO^* had much more TEAD1 bait than *Vgll4^fl/fl^* (Figure 5D,E), suggesting that the TEAD1–VGLL4 complex was disrupted in the *Vgll4^cKO^* hearts. As VGLL4 prevents the formation of the YAP–TEAD complex [18], we postulated that the TEAD1–YAP complex was enriched in *Vgll4^cKO^* hearts. In line with our expectation, YAP was detected in the TEAD1 pull-down product of *Vgll4^cKO^* hearts and was under the Western blotting detection threshold in the corresponding part of *Vgll4^fl/fl^* hearts (Figure 5D).

Together, these data confirmed that VGLL4 was efficiently depleted in the CMs of *Vgll4^cKO^* hearts.

### 3.6. CM-Specific Depletion of VGLL4 Does Not Affect Heart Growth and Function

We examined whether the loss of VGLL4 affects heart growth and function. Compared with *Vgll4^fl/fl^,* overall, *Vgll4^cKO^* mice had no growth defects, with their hearts showing normal size and morphology (Figure 6A,B). According to echocardiography, the heart function and wall thickness of adult *Vgll4^cKO^* mice were not distinguishable from those of littermate controls (Figure 6C). The histological analysis revealed that *Vgll4^cKO^* mice had normal ventricle wall thickness (Figure 6D,E) and regularly arranged cardiac myofibers (Figure 6F). Similar to their littermate controls, *Vgll4^cKO^* mice had minimal cardiac fibrosis (Appendix A). Together, these data suggest that the CM-specific depletion of VGLL4 does not affect heart growth and function.

### 3.7. CM-Specific Depletion of VGLL4 Does Not Affect CM Size

Although the CM-specific depletion of VGLL4 did not affect heart size, it might affect the CM size. We addressed this issue by first measuring the CM cross-sectional area in situ and found that the *Vgll4^cKO^* CM cross-sectional area was not significantly different from *Vgll4^fl/fl^* CMs at 2–3 months of age in female mice (Figure 7A,B). To further examine the CM size, we dissociated the hearts and measured the area, length, and width of isolated CMs. Consistent with the in situ measurement results, a two-dimensional examination of the isolated CMs revealed no significant difference between *Vgll4^fl/fl^* and *Vgll4^cKO^* CMs for both sexes at 2–3 months of age (Figure 7C–E). In the *Vgll4^cKO^* heart, the TEAD1 protein was increased, and more TEAD/YAP complex was formed (Figure 5D). To verify whether the loss of VGLL4 increases YAP transcriptional activity, we examined the expression of two YAP/TEAD1 target genes with the isolated CMs. The qRT-PCR results showed that the expression of *Ccn2* and not *Ccn1* was significantly increased in the *Vgll4^cKO^* CMs, thus suggesting that the loss of VGLL4 partially increases YAP transcriptional activity. We further examined whether the YAP expression and subcellular localization were altered in *Vgll4^cKO^* CMs, and the results showed that the YAP protein level and nuclei distribution were not distinguishable between the control and *Vgll4^cKO^* CMs (Appendix A).

Together with Figure 5 and Figure 6, these data indicate that VGLL4 is dispensable for myocardial growth and not required for maintaining the baseline heart function and that VGLL4 suppresses the YAP activity by disrupting the YAP/TEAD1 complex rather than affecting the YAP expression and subcellular distribution. Furthermore, by analyzing the cardiac phenotypes of both *Vgll4* d46 and *Vgll4^cKO^* mice, we concluded that, unlike Hippo kinases, the endogenous VGLL4 does not suppress YAP mitogenic activity in the fetal and neonatal cardiomyocytes (Figure 7G).

## 4. Discussion

In the past decade, mouse genetic studies have shown that the Hippo–YAP pathway controls fetal heart growth [10,12,13]. VGLL4 is a crucial suppressor of the YAP/TEAD1 complex [19], the terminal transcription regulator of the Hippo–YAP pathway. We previously showed that the activation of VGLL4 in the postnatal heart decreased CM proliferation and caused heart failure [21]. In the current study, we generated two mouse models to investigate the VGLL4 function in the heart. Although the whole-body deletion of VGLL4 resulted in perinatal lethality, the CM-specific depletion of VGLL4 did not cause observable heart growth defects, suggesting that VGLL4 is required for embryo development but is dispensable for myocardial growth.

In mice, the *Vgll4* gene has seven annotated transcript variants whose protein products are diverse in the N-termini and share the same sequences in the C-termini, and the alternative usage of the second exon is one of the mechanisms that govern the formation of these variants. Yu et al. previously reported that the whole-body depletion of *Vgll4* exon 2 (*Vgll4*^dE2^) caused pre-mature death and aortic valve defects [22]. Our current data show that the *Vgll4* X3 variant that does not contain exon 2 is highly expressed in the mouse heart, suggesting that the deletion of *Vgll4* exon 2 may not completely abolish the VGLL4 function. In this work, we generated a *Vgll4* mutant allele that disrupted TDU domains (*Vgll4*^d46^). Compared with the published *Vgll4*^dE2^ mutants [22], the phenotype of *Vgll4*^d46^ is more severe, as reflected by the fact that a fraction of the *Vgll4*^dE2^ mutants could survive up to 8 weeks, but none of the *Vgll4*^d46^ mutants could survive over 10 days. Additionally, the *Vgll4*^d46^ pups that survived for 10 days could not gain body weight. These observations suggest that the deletion of exon 2 cannot completely disrupt the VGLL4 function.

To examine whether the loss of VGLL4 affects heart development, we checked the heart structure with E16.5 embryos and found that the tricuspid valve was malformed, and the myocardium was normal in the *Vgll4*^d46^ mutants. The tricuspid valve growth defect was not observed in the *Vgll4*^dE2^ mutants, which displayed aortic valve and pulmonary valve malformation at both adult and neonatal stages [22]. Furthermore, although increased valve interstitial cell (VIC) proliferation was detected in the *Vgll4*^dE2^ mutants [22], the *Vgll4^d46^* tricuspid VICs had lower proliferation rates (Figure 4). The cardiac valve phenotype and VIC proliferation discrepancy between *Vgll4*^dE2^ and *Vgll4^d46^* may be due to the fact that the *Vgll4* gene has different transcript variants and that the *Vgll4* variants that do not contain exon 2 are not affected in *Vgll4*^dE2^, whereas all the *Vgll4* variants are disrupted in *Vgll4^d46^.*

The perinatal lethality of the *Vgll4*^d46^ mutants prevented us from studying the VGLL4 function in the postnatal heart; therefore, we generated a new *Vgll4* flox allele that has an exon 5 coding sequence flanked by LoxP sequences. By depleting VGLL4 in the CMs, we found that the loss of VGLL4 did not affect heart growth or heart function and that the CM size was not altered in these CM-specific *Vgll4* knockout (*Vgll4^cKO^*) mice. In addition to the fact that the *Vgll4^d46^* embryos had normal myocardial thicknesses and CM proliferation rates, the data from the adult *Vgll4^cKO^* mice further suggest that VGLL4 is dispensable for myocardial growth. Together, the heart morphology data collected from the *Vgll4*^dE2^, *Vgll4^d46^*, and *Vgll4^cKO^* mutants highlight the importance of VGLL4 in heart valve development and suggest that VGLL4 is not required for myocardial development.

As one of the essential terminal effectors of the Hippo–YAP integrative pathway, YAP senses developmental and stress signals and relays these signals to the nuclei to change the CMs’ gene expression, thereby controlling CM proliferation [12,30], survival [31], and homeostasis [32]. Since YAP is instrumental for heart development and homeostasis maintenance, its activity is tightly regulated by multiple suppressors and activators. As one of the categories of YAP suppressors, the components of the Hippo kinase cascade activate LATS1/2 kinases to phosphorylate and inactivate YAP [7]. The LATS1/2 restraining of the YAP activity is pivotal for heart development and cardiac homeostasis maintenance, because the genetic depletion of Salvador, an activator of LATS1/2, resulted in cardiac hyperplasia in the fetus [10] and expressing a LATS1/2-insensitive YAP-5SA mutant in the adult heart caused sudden death [33]. VGLL4 is another category of YAP suppressors, which suppresses YAP indirectly by engaging TEADs and preventing the formation of the YAP–TEAD complex [17].

We previously found that the primary interaction partner of TEAD1 is YAP in the neonatal heart and that overexpressing a K225R-mutated VGLL4 in the neonatal heart disrupted the TEAD1–YAP interaction, suppressed CM proliferation, and caused heart failure [21]. The current data further corroborate our previous observations that the TEAD1–YAP complex is the main driver of fetal and neonatal myocardial growth [12,21] and suggest that endogenous VGLL4 has a minimal role in the regulation of fetal or neonatal myocardial growth. In healthy adult CMs, YAP is primarily localized in the cytoplasm [34], and the major interaction partner of TEAD1 is VGLL4 instead of YAP [21]. Here, we found that knocking out VGLL4 in the adult CMs slightly increased the amount of the YAP–TEAD1 complex and unregulated its target gene, *Ctgf (Ccn2)*; however, the *Vgll4^cKO^* CMs had no noticeable phenotypes. These data suggest that the endogenous TEAD1–VGLL4 complex does not maintain adult CM homeostasis under baseline conditions. One possibility is that the Vestigial-like (VGLL) protein family members have redundant functions, and the other VGLL proteins may compensate for the VGLL4 function in the *Vgll4^KO^* CMs. The other possibility is that the loss of the cardiomyocyte VGLL4 is not sufficient to increase the amount of the YAP–TEAD complex to a phenotype-causing extent, and this is because most YAP proteins are localized in the cytoplasm of healthy adult CMs. Notably, all our studies were carried out in unstressed mice, and it remains unknown whether VGLL4 has significant pathophysiological roles in disease-stressed hearts, in which the LATS1/2 suppression of YAP is attenuated, and the nuclei YAP is enriched [31,35]. As YAP has a protective role in diseased hearts [31,36,37,38], future studies should investigate whether the loss of VGLL4 boosts the YAP activity and, therefore, protects the heart against pathological stresses.

One of the limits of the current study is that the primary pathophysiological reason leading to the *Vgll4^d46^* perinatal death was not clear. Since the deletion of *Vgll4* exon 2 in skeletal muscle progenitor cells causes skeletal muscle growth defects [39], it is possible that completely knocking out VGLL4 impairs the respiration muscles’ function, which results in the suffocation of the *Vgll4^d46^* mutant pups. The other limit is that we were not able to provide cardiac functional results to explain the consequences of tricuspid valve malformation caused by VGLL4 depletion. Further tissue-specific VGLL4 knockout studies are needed to dissect how VGLL4 regulates tricuspid valve development and function.

In summary, to address whether VGLL4 regulates heart development, we generated two VGLL4 loss-of-function mouse models. Our results suggest that VGLL4 regulates tricuspid development and does not contribute to myocardial growth. In addition, we confirmed the presence of multiple *Vgll4* transcript variants, and our results indicate that disrupting the VGLL4 TDU-domain coding sequence is an efficient way to completely abolish the VGLL4 function. Lastly, we found that the CM-specific depletion of VGLL4 facilitated the formation of the YAP/TEAD1 complex and increased the YAP target gene expression without changing the YAP protein level and subcellular distribution, further corroborating the current knowledge that VGLL4 suppresses the YAP activity by reducing TEAD availability. 

In conclusion, our data suggest that endogenous VGLL4 minimally regulates myocardial growth at the fetal and postnatal heart development stages and that Hippo kinases are the primary YAP restrainers during myocardial development (Figure 7G). Future studies are required to test whether VGLL4 regulates cardiac homeostasis under stress conditions.

## Figures and Tables

**Figure 1 cells-11-02832-f001:**
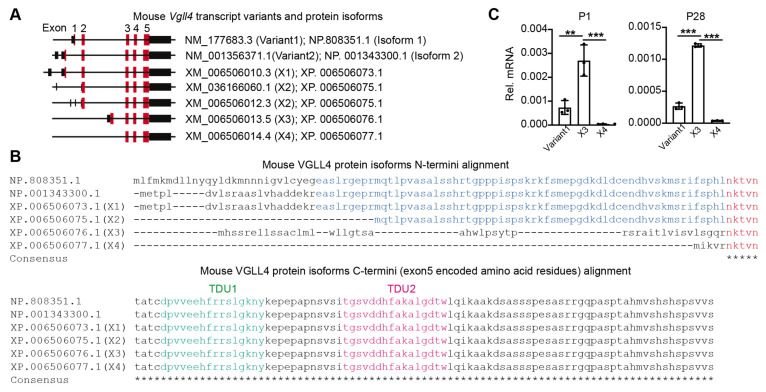
*Vgll4* transcript variants and their protein products: (**A**) schematic view of mouse *Vgll4* transcript variants. Black bars indicate non-translated mRNA sequences. Red bars indicate protein-coding sequences. The protein ID encoded by each variant was listed; (**B**) N-termini and C-termini alignments of mouse VGLL4 protein isoforms. N-termini alignment: blue and red letters indicate exon 2- and exon3-encoded amino acid residues, respectively. C-termini alignment: green and purple letters indicate TDU1 and TUD2 domains, respectively; (**C**) qRT-PCR measurement of three *Vgll4* transcript variants. Each dot indicates a data point from one individual animal. Total RNA from P1 and P28 hearts were used for qRT-PCR test. mRNA level of each isoform was normalized to *Gapdh*. One-way ANOVA test, ** *p* < 0.01, *** *p* < 0.001. N = 3.

**Figure 2 cells-11-02832-f002:**
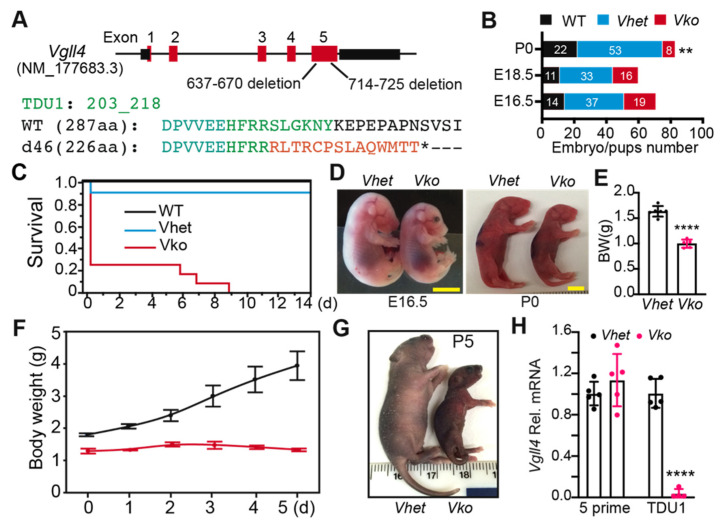
VGLL4 deletion resulted in perinatal lethality: (**A**) *Vgll4* gene structure showing *Vgll4*-mutant allele. Green and red letters indicate wild-type and d46 mutant amino acid sequences, respectively; * indicates a premature stop codon; (**B**) distribution of genotypes at E16.5, E18.5, and P0. Embryo/pup numbers for each genotype are displayed in the bar graph. **, Mendelian ratio chi-squared test, *p* < 0.01; (**C**) survival curve of *Vgll4 d46/+* (*Vhet*) and *Vgll4 d46/d46* (*Vko*) pups. Postnatal day 0 (P0) was designated as the date of delivery. For each group, N = 12; (**D**) gross morphology of embryos and mouse pups at indicated ages. Scale bar: 5 mm; (**E**) body weight at E18.5; (**F**) mouse pups’ body weight gain in the first 5 days after delivery. N = 3; (**G**) gross morphology of mouse pups at P5; (**H**) cardiac *Vgll4* mRNA expression. E13.5 hearts were collected for mRNA isolation and qRT-PCR measurement. TDU1 domain of *Vgll4* was not detectable in the *Vgll4* mutant transcripts. (**E**,**H**), Student’s *t*-test: **** *p* < 0.0001. N = 5–6.

**Figure 3 cells-11-02832-f003:**
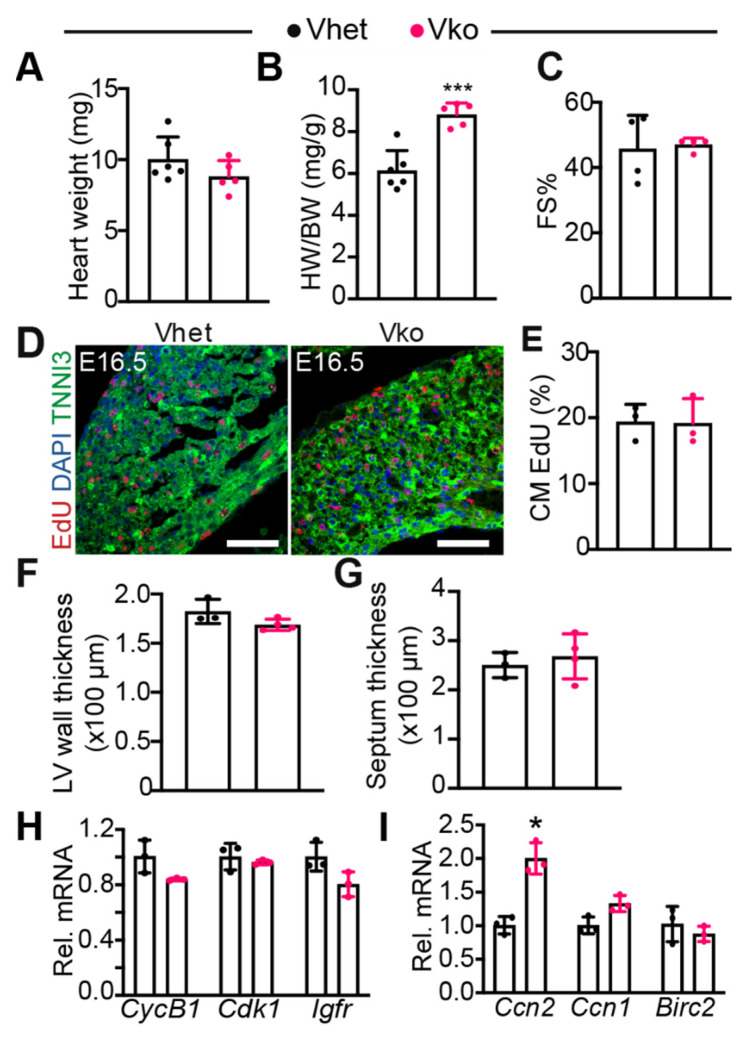
Loss of VGLL4 does not affect fetal myocardial growth: (**A**) heart weight; (**B**) heart-to-body weight ratio (**A**,**B**), *n* = 5–6; (**C**) echocardiography measurement of heart function at P0. FS%, fraction shortening. N = 4; (**D**) representative immunofluorescence images of E16.5 myocardium. Scale bar = 50 µm; (**E**) cardiomyocyte (CM) EdU incorporation rate. N = 3; (**F**,**G**) measurement of left ventricle free wall thickness (**F**) and septum thickness (**G**); (**H**,**I**) qRT-PCR measurement of cell proliferation-related genes (**H**) and YAP/TEAD target genes (**I**). Total RNA extracted from E16.5 myocardium was used for qRT-PCR. mRNA levels were normalized to *Gapdh*. Student’s *t*-test: * *p* < 0.05; *** *p* < 0.001. N = 3.

**Figure 4 cells-11-02832-f004:**
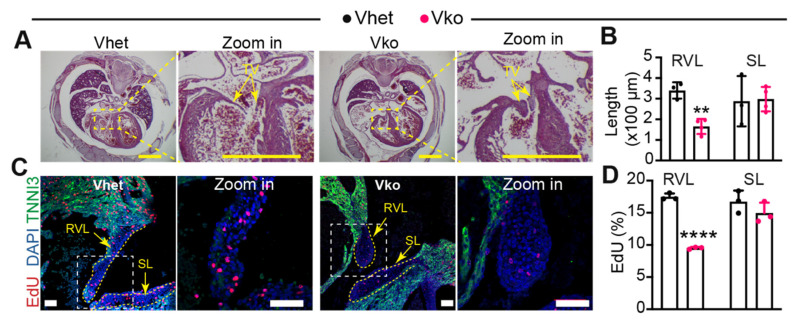
Loss of VGLL4 causes tricuspid valve malformation: (**A**) histology of embryonic heart at E16.5. TV, tricuspid valve. Scale bar = 1 mm; (**B**) measurement of right ventricle side (RV) tricuspid valve leaflet length and septal leaflet length. N = 3–4; (**C**) representative immunofluorescence images of tricuspid valve at E16.5. Scale bar = 50 µm. RVL, right ventricle side tricuspid valve leaflet. SL, septal leaflet; (**D**) EdU incorporation rate of tricuspid valve interstitial cells. (**B**,**D**), Student’s *t*-test: ** *p* < 0.01; **** *p* < 0.0001. N = 3. A and C, arrows indicate tricuspid valve leaflets.

**Figure 5 cells-11-02832-f005:**
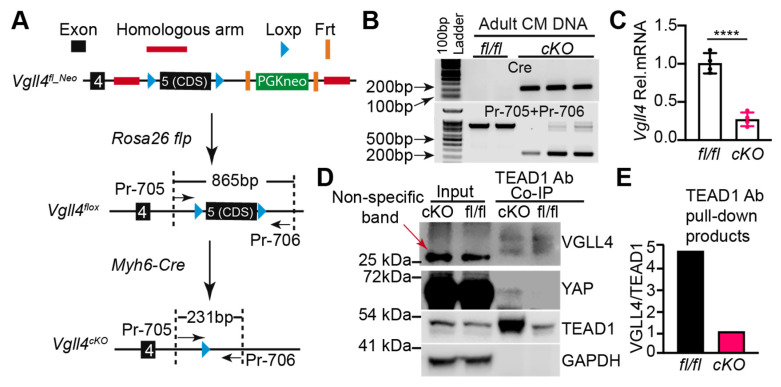
Generation and validation of *a Vgll4 flox* allele: (**A**)cardiac-specific *Vgll4* depletion strategy. The coding sequence of exon 5 (exon 5 CDS) was flanked by LoxP sites. A PGKneo selection cassette flanked by Frt sites was used for selection of recombinants. F1 *Vgll4^flox_Neo^* mice were bred to Rosa26 flp mice to remove the PGKneo cassette. Crossing *Vgll4 flox* allele with Myh6-Cre depleted exon 5 CDS in the CMs; (**B**) PCR validation of *Vgll4* exon 5 CDS excision; (**C**) qRT-PCR measurement of *Vgll4* mRNA. Student’s *t*-test, **** *p* < 0.0001, N = 4; (**B**,**C**) adult hearts were dissociated to yield purified CMs for DNA (**B**) and RNA (**C**) isolation; (**D**) co-immunoprecipitation (Co-IP) assay. Mouse origin TEAD1 antibody (BD, 610923) was used to pull down VGLL4 from heart lysate, and Rabbit origin TEAD1 antibody (CST, 12292S) was used in immunoblot to detect TEAD1. Red arrow indicates a non-specific band. GAPDH was used as loading control; (**E**) VGLL4 and TEAD1 protein ratio, determined by quantification of (**D**). Precipitated VGLL4 proteins were normalized to TEAD1.

**Figure 6 cells-11-02832-f006:**
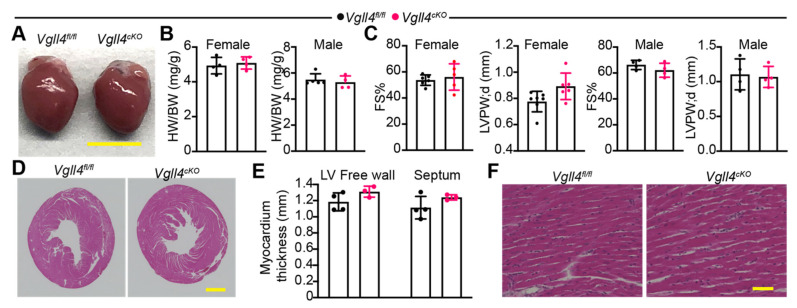
CM-specific depletion of VGLL4 does not affect myocardial growth and heart function: (**A**) gross morphology of control (*Vgll4^fl/fl^*) and *Myh6-Cre; Vgll4^fl/fl^* (*Vgll4^cKO^*) hearts at 8 weeks after birth. Scale bar = 5 mm; (**B**) heart-to-body weight ratio. N= 4–5; (**C**) echocardiography measurement of heart function and ventricle wall thickness. FS%, fraction shortening. LVPW, d: left ventricle posterior wall thickness; diastolic. N = 5–6; (**D**) H&E-stained heart cross-sections. Scale bar = 1 mm; (**E**) left ventricle wall and septum thickness; (**F**) longitudinal view of CM alignment. Scale bar = 50 µm.

**Figure 7 cells-11-02832-f007:**
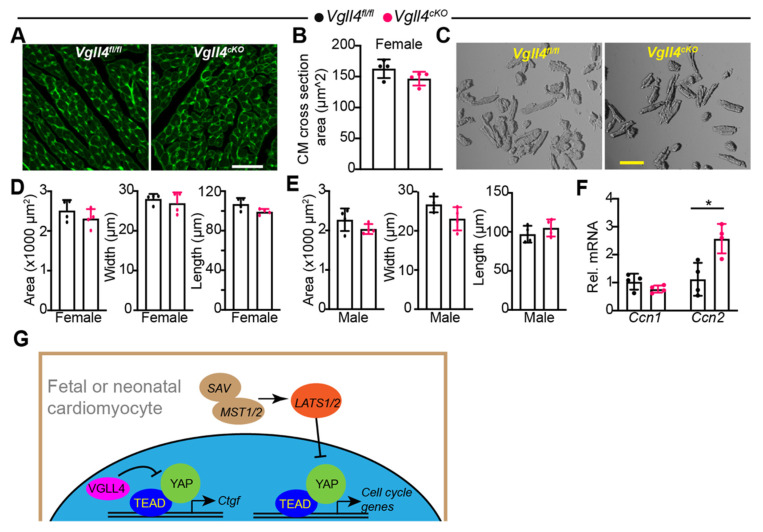
CM-specific depletion of VGLL4 does not affect CM size: (**A**) wheat germ agglutinin (WGA)-stained myocardium. Scale bar = 50 µm; (**B**) CM cross-sectional area. In each heart, 100–150 CMs were measured, and their average values are presented. A and B, heart sections from female mice were used. For each group, N = 4; (**C**) bright-field images of isolated CMs. Scale bar = 100 µm; (**D**,**E**) CM area, width, and length. Both female (**D**) and male (**E**) mice were included in this analysis. Average values of 150 isolated CMs from each dissociated heart are presented. For each group, N = 4; (**F**) qRT-PCR measurement of *Ccn1* (*Cyr61*) and *Ccn2* (*Ctgf*). Total RNA extracted from isolated CMs was used for qRT-PCR. mRNA levels were normalized to *Gapdh*. Student’s *t*-test: * *p* < 0.05. N = 4; (**G**) diagram summary of the main findings of this study. Arrows indicate activation. Blunt-ended lines indicate suppression.

## Data Availability

Mouse *Vgll4* transcript variants annotation can be retrieved from NCBI (https://www.ncbi.nlm.nih.gov/gene/232334).

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
