# Peer review of "Depletion of VGLL4 Causes Perinatal Lethality without Affecting Myocardial Development"

_cells, 2022, doi:10.3390/cells11182832_

Round 1
Reviewer 1 Report
Concerns:
1. Figure 1c, which isoform is dominant in mouse cardiomyocyte specifically?
2. Would be great if statisticals analysis can be performed for figure 2B.
3. Figure 3 B, unbalanced growth of the heart size can be considered a defect, same pumping capacity and cardiac output might cause cardiovascular defects, can the author exam or comment on potential cardiovascular defects such as abnormal blood pressure. While the whole body is under-developed, the heart seems to be over-developed, should that be considered as a phenotype?
4. Figure 4, Vko seems to have myxomatous TV, an alcian blue staining will reveal more info about proteoglycan level etc. It would also be informative to perform a pentachrome to examine the stratified layer structure starting to form at E16.5. Can the authors perform echocardiography to inspect valvular function such as regurgitation.
5. Did VGLL4 deletion in cardiomyocytes increase YAP activity? Such as expression level, nuclei localization, expression level of Yap target genes.
Author Response
We thank the reviewers for giving us valuable and constructive comments and suggestions. Here are our point-by-point responses.
Concerns:
- Figure 1c, which isoform is dominant in mouse cardiomyocyte specifically?
Response: Based on the qRT-PCR data, Vgll4 variant 1 and x3 have highest expression in the heart. However, we do not have good antibodies to confirm which VGLL4 protein isoform has the highest expression in the heart. Therefore, at the current stage, we are not able to determine which VGLL4 isoform is the dominant one in the cardiomyocyte.
- Would be great if statisticals analysis can be performed for figure 2B.
Response: We displayed the exact embryo/pups number for each genotype, and performed Chi-squared test with the genotype distribution of embryos/pups. The P value of P0 pups is 0.0039, which is significant different from the expected Mendelian ratio. We updated Figure 2 and its legend.
- Figure 3 B, unbalanced growth of the heart size can be considered a defect, same pumping capacity and cardiac output might cause cardiovascular defects, can the author exam or comment on potential cardiovascular defects such as abnormal blood pressure. While the whole body is under-developed, the heart seems to be over-developed, should that be considered as a phenotype?
Response: Our previous work has shown that VGLL4 has minimum expression in the neonatal heart [1], suggesting that endogenous VGLL4 might not regulate YAP activity in the fetal or neonatal myocardium. In support of this view, our data show that the VGLL4 KO myocardium is not over-developed (Fig. 3 D, E, F and G). Rather, we believe that VGLL4 is required for the development of other tissues, such as skeletal muscle. This opinion is supported by a recent report, which showed that knocking down VGLL4 decreased the skeletal muscle myofiber size [2]. In summary, it is likely that loss of VGLL4 disrupts body development without affecting myocardial growth.
Our echo data showed that the P1 Vgll4 KO pups had normal systolic heart function (Fig. 3C). We did not examine blood pressure due to technical difficulties and the scarce of Vgll4 KO pups. Nevertheless, when we knocked out VGLL4 only in the cardiomyocytes, the adult conditional Vgll4 knockout mice had normal heart morphology and function (Fig. 6), further supporting our view that VGLL4 is dispensable for myocardial growth.
- Figure 4, Vko seems to have myxomatous TV, an alcian blue staining will reveal more info about proteoglycan level etc. It would also be informative to perform a pentachrome to examine the stratified layer structure starting to form at E16.5. Can the authors perform echocardiography to inspect valvular function such as regurgitation.
Response: The reviewer's suggestions are very constructive. However, we were not able to perform these experiments in the current lab. First, we did not keep the Vgll4 d46 mouse line. Second, the current lab does not have the expertise of performing echo measurements with E16.5 embryos. With the newly generated VGLL4 flox allele in hand, we hope that we can find some collaborators in the future to address these intriguing questions, and to determine how VGLL4 regulates tricuspid valve development.
- Did VGLL4 deletion in cardiomyocytes increase YAP activity? Such as expression level, nuclei localization, expression level of Yap target genes.
Response: VGLL4 suppresses YAP by competing for TEADs. We found that TEAD1 pulled down YAP in the adult VGLL4 cKO heart, but not in the littermate control Vgll4 fl/fl heart (Fig. 5D), suggesting that YAP activity is increased in the Vgll4 cKO heart. In support of this hypothesis, depletion of VGLL4 increased the expression of YAP target gene Ccn2 (also known as Ctgf) (Fig. 3I; Fig. 7F). We also examined YAP protein expression and nuclei distribution in both control and Vgll4 cKO hearts. Our results showed that YAP expression and sub-cellular distribution was not affected in VGLL4 cKO cardiomyocytes (See newly added Suppl. Fig. 6). These data together suggest that VGLL4 suppresses YAP activity by disrupting YAP/TEAD1 complex rather than affecting YAP expression and sub-cellular distribution.
References:
[1] Lin, Z., Guo, H., Cao, Y., Zohrabian, S., Zhou, P., Ma, Q., VanDusen, N., Guo, Y., Zhang, J., Stevens, S.M., Liang, F., Quan, Q., van Gorp, P.R., Li, A., Dos Remedios, C., He, A., Bezzerides, V.J. and Pu, W.T. (2016) Acetylation of VGLL4 Regulates Hippo-YAP Signaling and Postnatal Cardiac Growth. Dev Cell 39, 466-479.
[2] Feng, X., Wang, Z., Wang, F., Lu, T., Xu, J., Ma, X., Li, J., He, L., Zhang, W., Li, S., Yang, W., Zhang, S., Ge, G., Zhao, Y., Hu, P. and Zhang, L. (2019) Dual function of VGLL4 in muscle regeneration. EMBO J 38, e101051.
Reviewer 2 Report
The manuscript describes an important pathway involved in the myocardial development. Overall this is a concise and well written article with clear results. However, some clarification is needed in regards to some experiments and texts. In my point of view, the authors should address the following points -
1. To give general readership a reasonable background, the authors should consider including a separate paragraph in the introduction section about Hippo-YAP pathway and LATS1/2 activity with respect to neonatal and adult heart development.
2. Similarly, the authors should provide the DNA alignment sequence for Rosa26 flp and Myh6-cre transgenic mice used for generation of cardiac-specific Vgll4 knockout mouse line.
3. Do the authors obtained any functional data for the hearts (Echo)? It would be nice to assess the functionality. Please add this to show tricuspid valve dysfunction and other issues.
4. How healthy are the “mitochondria” present in the KO tissue? Have authors assessed this?
5. Have the authors check for fibrosis in the tissue section?
Author Response
The manuscript describes an important pathway involved in the myocardial development. Overall this is a concise and well written article with clear results. However, some clarification is needed in regards to some experiments and texts. In my point of view, the authors should address the following points -
Response: We thank the reviewer for the nice comments and constructive suggestions. The following paragraphs are our point-to-point responses.
- To give general readership a reasonable background, the authors should consider including a separate paragraph in the introduction section about Hippo-YAP pathway and LATS1/2 activity with respect to neonatal and adult heart development.
Response: In the introduction section, we added one paragraph to describe the roles of crucial Hippo kinases in heart development.
- Similarly, the authors should provide the DNA alignment sequence for Rosa26 flp and Myh6-cre transgenic mice used for generation of cardiac-specific Vgll4 knockout mouse line.
Response: We are not very clear about this request. We assumed that the reviewer asked us to add some background introduction to these two transgenic mouse lines. On page 8, we added several sentences to introduce these two mouse lines.
- Do the authors obtained any functional data for the hearts (Echo)? It would be nice to assess the functionality. Please add this to show tricuspid valve dysfunction and other issues.
Response: We performed echo measurements with the P1 Vko hearts (Fig. 3C) and adult Vgll4 cKO mice (Fig. 6C). Unfortunately, we do not have the expertise of performing Doppler echocardiography with E16.5 embryos, and so we were not able to measure the tricuspid valve function. With the newly generated VGLL4 flox allele in hand, we hope that we can find some collaborators in the future to determine whether and how VGLL4 regulates tricuspid valve development and function.
- How healthy are the “mitochondria” present in the KO tissue? Have authors assessed this?
Response: It a good suggestion to check the mitochondria in Vgll4 cKO cardiomyocytes. We are now characterizing the mitochondria morphology. The Vgll4 cKO cardiomyocytes seem to have smaller mitochondria; however, the data are not conclusive. Therefore, we would rather not include the mitochondria data in this manuscript.
- Have the authors check for fibrosis in the tissue section?
Response: We did Trichrome staining with the adult control and Vgll4 cKO hearts. Same with the littermate control heart, Vgll4 cKO heart did not have obvious fibrosis (See newly added Suppl. Fig. 5).
Round 2
Reviewer 2 Report
Accepted
Author Response
Thank you.